REGISTERED REPORT PROTOCOL

# Protocol for educational programs on infection prevention/control for medical and healthcare student: A systematic review and meta-analysis

Akira Yoshikawa[ID][1], Naonori Tashiro[ID][2,3]*, Hiroyuki Ohtsuka[2], Keiichiro Aoki[4], Shusuke Togo[5,6], Kazuki Komaba[4,7], Satoshi Nogawa[1,8], Miwa Osawa[1,9], Megumi Enokida[1,5]

1 Division of Health Science Education, Showa University School of Nursing and Rehabilitation Sciences, Yokohama, Kanagawa, Japan, 2 Department of Physical Therapy, Showa University School of Nursing and Rehabilitation Sciences, Yokohama, Kanagawa, Japan, 3 Rehabilitation Center, Showa University Hospital, Tokyo, Japan, 4 Department of Occupational Therapy, Showa University School of Nursing and Rehabilitation Sciences, Yokohama, Kanagawa, Japan, 5 Department of Nursing, Showa University School of Nursing and Rehabilitation Sciences, Yokohama, Kanagawa, Japan, 6 Department of Nursing, Showa University Fujigaoka Hospital, Yokohama, Kanagawa, Japan, 7 Rehabilitation Center, Showa University Fujigaoka Hospital, Yokohama, Kanagawa, Japan, 8 Department of Clinical Engineering, Showa University Fujigaoka Hospital, Yokohama, Kanagawa, Japan, 9 Department of Radiological Technology, Showa University Fujigaoka Hospital, Yokohama, Kanagawa, Japan

* tashiro@cmed.showa-u.ac.jp

## Abstract

During the COVID-19 pandemic, infection protection/control education has become increasingly important for not only healthcare professionals but also students undertaking medical, nursing, physical therapy, occupational therapy, and other related courses. A review of the literature on infection control education reveals that the target participants often comprise healthcare workers, and very few studies of infection control education focus on students. We have developed a protocol for the systematic review of the literature on simulation-based infection prevention/control education for students undertaking medical, nursing, rehabilitation, and other related courses. The protocol for the systematic review and meta-analysis has been drafted in alignment with the Preferred Reporting Items for Systematic Reviews and Meta-Analysis (PRISMA) statement. Systematic literature search will be performed for the period between 1990 (January) and 2022 (September) using the CENTRAL, MEDLINE, and Scopus databases. We will qualitatively and quantitatively examine the effects of simulation-based infection education for students in this systematic review and meta-analysis. Two investigators will independently search the databases according to the defined search strategy. The full-text of the selected articles will be screened independently keeping in mind the inclusion criteria by a pair of reviewers. Descriptive data will be extracted from each study regarding: study design, methods, participants, and outcomes. A meta-analysis will be performed if the quantitative data is suitable. Heterogeneity will be assessed using the standard $\chi^2$. Odds ratio for categorical data and weighted mean differences for continuous data and their 95% confidence intervals will be calculated and used for analysis. Where statistical pooling is not possible, the findings of the quantitative papers will

**Data Availability Statement:** All relevant data from this study will be made available upon study completion.

**Funding:** This study was supported by School of Nursing and Rehabilitation Sciences Showa University Research Fund (Grant Numbers 2022No.3: AY). The funders had no role in the study design, data collection and analysis, decision to publish, or preparation of the manuscript. There was no additional external funding received for this study.

**Competing interests:** The authors have declared that no competing interests exist.

be presented in narrative form. The qualitative aspect will employ narrative (descriptive) synthesis. Our review will make a valuable contribution to the domain of simulation-based infection prevention/control for students enrolled in medical and/or related courses.

## Introduction

Maintaining hand hygiene is an infection prevention measure that everyone can easily practice. The World Health Organization (WHO) recommends practicing hand hygiene as an infection control measure to protect ourselves from the effects of the coronavirus disease 2019 (COVID-19) pandemic [1]. Under these circumstances, infection protection/control education has become increasingly important for not only healthcare professionals but also students enrolled in the medical, nursing, physical therapy, occupational therapy, and other related school courses. A review of the literature on infection control education (lectures on infectiology, how to practice hand hygiene, and/or how to use personal protective equipment, etc.) reveals that the subjects are often healthcare professionals [2–6]. However, there are very few studies on infection control education focused on students, and the development of structured infection control programs is still underway. Therefore, evidence of the effects of such education on students is limited and inconclusive [7–9]. Recently, it was reported that though students were able to comprehend information regarding infection protection/control of the current COVID-19 pandemic, they did not attain the skill level to practice infection protection/control [10]. Thus, students suffer from fear of infection [10,11]. It also became clear that students harbored a strong desire to contain the spread of COVID-19 through their own actions [11]. These factors suggest the need to establish systematic infection protection/control education programs for students and clarify their effectiveness.

Simulation, role-play, skill training, electronic learning, and face-to-face lectures are methods of infection protection/control education for students. A study on infection education reported that simulation education using a standardized patient was a significantly more effective method than role-play for nursing students [12]. The effectiveness of simulation education on infection control for students is otherwise not fully known. Instead, simulation education on infection control is focused on prevention of healthcare-associated infections (HAIs) for healthcare professionals. HAIs include catheter-associated bloodstream infections, catheter-associated urinary tract infections, surgical site infections, and ventilator-associated pneumonia; 60–70% of these infections can be prevented [3,4,13]. Therefore, it can be inferred that simulation education on infection protection/control has been provided for healthcare professionals. However, simulation education focused on students employed several methods comprising low-tech simulators to high fidelity simulators to improve their ability: in knowledge, critical thinking ability, satisfaction, or confidence [14,15]. Systematic reviews reported that simulation education for students belonging to medical, nursing, and physical therapy courses significantly improved their learning in terms of knowledge retention, clinical thinking, practical skills, confidence, and satisfaction compared to traditional learning methods [15–18]. Therefore, it is very important to establish a simulation-based education on infection prevention/control for students and to verify and clarify the effectiveness of such education.

Our systematic review and meta-analysis' research question is: What enhances learning satisfaction and the efficacy of simulation-based infection prevention/control education, compared to classical education, among students who are enrolled in medical, nursing, rehabilitation, and other related courses? Therefore, we will search for literature published

between 1990 (January) and 2022 (December) using the CENTRAL, MEDLINE, and Scopus databases. Following which, we will qualitatively and quantitatively examine the effects of simulation-based infection education on students in this systematic review and meta-analysis. Our review will make a valuable contribution to the domain of simulation-based infection prevention/control for students enrolled in medical and/or related courses, and will help establish educational programs for infection prevention/control.

## Materials and methods

This systematic review protocol was preregistered in the Open Science Framework Registries: https://osf.io/q27cj/. The present study protocol is being reported in accordance with the reporting guidance provided in the PRISMA Protocols (PRISMA-P) statement [19].

### Inclusion criteria

The studies will be selected according to the following criteria:

**Types of studies.**   We will include all controlled clinical trial and randomized controlled trials (RCTs). Further, we will supplement these with observational studies (including cohort and case-control studies) to obtain results of the practical reports.

**Types of participants.**   The included studies will comprise participants who are undergraduate and graduate students enrolled in medical and healthcare-related occupational courses (medicine, dentistry, nursing, physical therapy, occupational therapy, pharmacy, and other healthcare-related fields).

**Types of outcome measures.**   The following outcome measures will be considered while including studies: critical thinking, skill performance, knowledge acquisition, decision making and problem-solving skills, self-efficacy, clinical reasoning skills, self-confidence, communication skills, teamwork, improved clinical performance, leadership skills, and student satisfaction.

### Exclusion criteria

This study is a systematic review of the simulation-based infection prevention/control education for students. Therefore, studies involving healthcare professionals or non-university students (e.g., kindergarten and elementary school students) will be excluded.

### Search methods for identifying studies

We will use a combination of text words and medical subject headings (MeSH) terms depending on the database to capture the following concepts: effectiveness of infection prevention/control education and intervention as the simulation-based education. The terms used will be: "students," "health occupations," "pupil nurse," "simulation training," "infection," "randomized," "clinical trials." Comprehensive searches will be conducted in the CENTRAL, MEDLINE, and Scopus databases.

### Data collection and analysis

**Selection of studies.**   First, two investigators will screen titles and abstracts using the text words and MeSH terms outlined previously in the initial literature search to determine whether articles potentially meet the inclusion criteria; articles that clearly do not meet the criteria will be rejected. In this primary screening phase, articles that do not match the review question are excluded by analyzing the title and abstract, and those that cannot be judged from the abstract are retained in principle. Second, the two reviewers will review the full text of the

remaining articles independently to determine their eligibility in the review process. If it is not possible to extract all necessary results of the primary, secondary, and other outcome from an included study, attempts will be made to contact authors to account for any missing data in the studies. If the study authors do not respond or if the data is unavailable, this will be mentioned in the report and the data will be presented in the supplementary information. Disagreements at any stage will be resolved through discussion between the two reviewers. If the reviewers fail to reach a consensus, a third reviewer will be consulted for arbitration.

**Data extraction and management.**   The data extraction sheet will be piloted among the reviewers before extraction begins—to ensure that it is easy to use. in eight reviewers. After this, data extraction will be conducted by two reviewers independently, recorded and managed using standard Microsoft Excel data recording spreadsheets by eight reviewers. Data will be extracted to obtain a complete record of the methodology, study design, participants, interventions, outcome measures, and results. Maximal data extraction is planned to ensure that findings can be adequately followed up without returning to the original data set. Data to be extracted conforms with the Cochrane recommendations.

**Assessment of risk of bias of included studies.**   To assess the possible risk of bias for each study, we will evaluate and report on the methodological risk of bias for the included studies on the following individual elements for RCTs: random sequence generation, allocation sequence concealment, blinding (participants, personnel), blinding (outcome assessment), completeness of outcome data, selective outcome reporting, similar baseline characteristics, and similar baseline outcome measurements.

In all cases, two reviewers will independently assess the risk of bias for the included studies, with any disagreements resolved through discussion or by consulting a third reviewer who was expected to be consulted previously for arbitration till a consensus is reached. We will judge each item as being at high, low, or unclear risk of bias as set out in the criteria. We will contact study authors for additional information about the included studies, or for clarification of the study methods as required.

## Data synthesis and statistical analysis

We will perform a meta-analysis to assess the included studies' clinical and methodological diversity and statistical heterogeneity. For continuous outcomes, we will use the mean difference or standardized mean difference, as appropriate. In addition, for dichotomous outcomes, we will adopt the risk difference or risk ratio, as appropriate. Where quantitative integration is not possible, the results will be analyzed and described. If we can statistically pool the results, we will provide forest plots to summarize the results of individual studies. For data synthesis, we will use the Review Manager software, version 5.4 (Cochrane Collaboration, Oxford, UK).

We intend to use the Granding of Recommendations Assessment, Development and Evaluation (GRADE) approach to assess the overall strength of the evidence assessment. In RCT, the GRADE approach evaluates the limitations of the study, inconsistencies, indirect evidence, inaccuracies, and publication bias, and classifies the evidence as high, moderate, low, or very low [20]. Where statistical pooling is not possible, the findings will be presented in a narrative form including tables and figures on content analysis of findings to aid in data presentation, as and where appropriate. The review will seek to synthesize the quantitative simulation-based infection prevention/control articles included through pooled statistical meta-analysis, if sufficiently homogenous articles are retrieved (at least three homogenous articles). Heterogeneity will be assessed using the standard $\chi^2$. Odds ratio for categorical data and weighted mean differences for continuous data and their 95% confidence intervals will be calculated and used for the analysis. Where statistical pooling is not possible, the findings from the quantitative papers

will be presented in the narrative form. The qualitative aspect will employ narrative (descriptive) synthesis. Quantitative, qualitative, and mixed methods studies selected for retrieval will be synthesized by two reviewers who reviewed the paper for methodological validity prior to inclusion in the review. Any disagreement that may arise between the reviewers will be resolved through discussion or with a third reviewer until a consensus is reached.

## Status of the study

The study is in the data collection and analysis phase. The initial deadline for completion of the same is July 31, 2023.

## Discussion

Infection prevention/control education has a wide range of benefits, including protecting healthcare professionals themselves, controlling nosocomial infections, reducing the infection rate, and reducing the medical cost related the infection [3,4,21]. During the COVID-19 pandemic, the WHO published guidelines on occupational safety for healthcare professionals since they are in the front line of any outbreak [22]. WHO recommends providing adequate training, infection prevention and control education and personal protective equipment for the occupational safety of healthcare professionals [22]. Despite this, unfortunately, COVID-19 infections have claimed the lives of many healthcare professionals [23]. To prevent a repetition of this tragedy, it is very important to enhance infection prevention/control education for students who want to become healthcare professionals. Most studies on infection control education are for healthcare professionals. Simulation-based educational methods are beginning to be implemented for infection control education. These are also educational studies targeting healthcare professionals, and several aspects regarding their effectiveness in teaching students remain unknown. To resolve the same, a critical evaluation and comprehensive synthesis of the available evidence will be performed in the systematic review to assess the efficacy of simulation-based infection prevention/control education for students. Hopefully, conclusions drawn from the review will benefit students enrolled in medical, nursing, physical therapy, occupational therapy, and other related courses and lecturers of medical courses in the concerned colleges or universities. The process of conducting this systematic review will include selection and inclusion of studies, data extraction, and data synthesis. If amendments are necessary, the date and statement of changes with their corresponding reasons will be provided.

## Acknowledgments

We would like to thank Ms. Tomoko Morimasa and Ms. Asae Ito (librarian, Showa university) for their advice in creating the medical subject headings (MeSH) terms. We would like to thank Ms. Marina Fukao (the staff of Showa university) for administrative and clerical support. Finally, we would like to thank Editage (www.editage.com) for English language editing.

## Author Contributions

**Conceptualization:** Akira Yoshikawa, Naonori Tashiro, Megumi Enokida.

**Data curation:** Akira Yoshikawa, Naonori Tashiro, Hiroyuki Ohtsuka, Keiichiro Aoki.

**Formal analysis:** Akira Yoshikawa, Naonori Tashiro, Hiroyuki Ohtsuka, Keiichiro Aoki.

**Funding acquisition:** Akira Yoshikawa.

**Investigation:** Akira Yoshikawa, Naonori Tashiro, Hiroyuki Ohtsuka, Keiichiro Aoki, Shusuke Togo, Kazuki Komaba, Satoshi Nogawa, Miwa Osawa.

**Methodology:** Akira Yoshikawa, Naonori Tashiro, Hiroyuki Ohtsuka, Keiichiro Aoki.

**Project administration:** Akira Yoshikawa, Naonori Tashiro, Megumi Enokida.

**Resources:** Akira Yoshikawa, Naonori Tashiro, Hiroyuki Ohtsuka, Keiichiro Aoki, Shusuke Togo, Kazuki Komaba, Satoshi Nogawa, Miwa Osawa.

**Software:** Naonori Tashiro, Hiroyuki Ohtsuka.

**Supervision:** Naonori Tashiro, Megumi Enokida.

**Validation:** Akira Yoshikawa, Naonori Tashiro, Hiroyuki Ohtsuka, Keiichiro Aoki, Shusuke Togo, Kazuki Komaba, Satoshi Nogawa, Miwa Osawa.

**Visualization:** Akira Yoshikawa, Naonori Tashiro.

**Writing – original draft:** Akira Yoshikawa.

**Writing – review & editing:** Akira Yoshikawa, Naonori Tashiro, Megumi Enokida.

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
