## [Decision Letter · Decision Letter 0]

6 Sep 2022

PONE-D-22-19321Systematic review protocol for educational program on infection prevention/control for students in medical and healthcare schoolsPLOS ONE

Dear Dr. Tashiro,

Thank you for submitting your manuscript to PLOS ONE. After careful consideration, we feel that it has merit but does not fully meet PLOS ONE’s publication criteria as it currently stands. Therefore, we invite you to submit a revised version of the manuscript that addresses the points raised during the review process.

We look forward to receiving your revised manuscript.

Kind regards,

Sawsan Abuhammad

Academic Editor

PLOS ONE

Journal Requirements:

"This study was supported by School of Nursing and Rehabilitation Sciences Showa University Research Fund (Grant Numbers 2022No.3: AY)."

"This study was supported by School of Nursing and Rehabilitation Sciences Showa University Research Fund (Grant Numbers 2022No.3: AY)."

"The authors declare no competing interests."

Reviewers' comments:

Reviewer's Responses to Questions

**Comments to the Author**

1. Does the manuscript provide a valid rationale for the proposed study, with clearly identified and justified research questions?

Reviewer #1: Partly

2. Is the protocol technically sound and planned in a manner that will lead to a meaningful outcome and allow testing the stated hypotheses?

Reviewer #1: Partly

3. Is the methodology feasible and described in sufficient detail to allow the work to be replicable?

Reviewer #1: No

4. Have the authors described where all data underlying the findings will be made available when the study is complete?

Reviewer #1: Yes

5. Is the manuscript presented in an intelligible fashion and written in standard English?

Reviewer #1: Yes

6. Review Comments to the Author

You may also provide optional suggestions and comments to authors that they might find helpful in planning their study.

Reviewer #1: • The research question needs to be clearly defined.

• Literature search and study selection: Just two bibliographic databases were included. Typically, three bibliographic databases should be used. In order to secure proper basis for evidence-based research, it is essential to perform a broad search that includes as many studies as possible that meet the inclusion and exclusion criteria.

• Inclusion criteria/ Types of participants: I think the inclusion of studies has biases. It is not comprehensive. Graduate students not included. However, many studies conducted on graduate students with simulation lab. Also, why medicine students and pharmacy students not included, this could yield misleading results.

• Guidelines for reporting meta-analyses of RCTs were not presented clearly.

7. PLOS authors have the option to publish the peer review history of their article (what does this mean?). If published, this will include your full peer review and any attached files.

Reviewer #1: **Yes: **Ala'a Fawwaz Dalky

---

## [Author Response · Author response to Decision Letter 0]

11 Oct 2022

Response to Journal Requirements:

Thank you very much for providing important comments. We are thankful for the time and energy you expended. Our responses to the referees’ comments are as follow:

RESPONSE: Thank you for pointing this out. We have formatted the manuscript in accordance with the journal guidelines.

2. Thank you for stating in your Funding Statement: "This study was supported by School of Nursing and Rehabilitation Sciences Showa University Research Fund (Grant Numbers 2022No.3: AY)."

Please provide an amended statement that declares *all* the funding or sources of support (whether external or internal to your organization) received during this study, as detailed online in our guide for authors at http://journals.plos.org/plosone/s/submit-now. Please also include the statement “There was no additional external funding received for this study.” in your updated Funding Statement. Please include your amended Funding Statement within your cover letter. We will change the online submission form on your behalf.

RESPONSE: Thank you for pointing this out. This study received no additional external funding, therefore, we have added a sentence regarding this (line 284).

3. Thank you for stating in your Funding Statement: "This study was supported by School of Nursing and Rehabilitation Sciences Showa University Research Fund (Grant Numbers 2022No.3: AY)."

Please state what role the funders took in the study. If the funders had no role, please state: ""The funders had no role in study design, data collection and analysis, decision to publish, or preparation of the manuscript."" If this statement is not correct you must amend it as needed. Please include this amended Role of Funder statement in your cover letter; we will change the online submission form on your behalf.

RESPONSE: Thank you for pointing this out. The funders had no role in the study design, data collection and analysis, decision to publish, or preparation of the manuscript; therefore, per your comment, we have stated this in the manuscript (lines 282–284) and covering letter.

4. Thank you for stating the following in your Competing Interests section: "The authors declare no competing interests."

Please complete your Competing Interests on the online submission form to state any Competing Interests. If you have no competing interests, please state ""The authors have declared that no competing interests exist."", as detailed online in our guide for authors at http://journals.plos.org/plosone/s/submit-now This information should be included in your cover letter; we will change the online submission form on your behalf.

RESPONSE: Thank you for pointing this out. Per your comment, we have included a sentence declaring that no competing interests exist in the cover letter document labeled “Revised cover letter.”

RESPONSE: Thank you for pointing this out. We have re-checked the reference list and the text.

 

Response to Reviewer #1

We are grateful for this kind comment. Our responses to the referees’ comments are as follow:

1. Does the manuscript provide a valid rationale for the proposed study, with clearly identified and justified research questions?

Reviewer #1: Partly

2. Is the protocol technically sound and planned in a manner that will lead to a meaningful outcome and allow testing the stated hypotheses?

Reviewer #1: Partly

3. Is the methodology feasible and described in sufficient detail to allow the work to be replicable?

Reviewer #1: No

RESPONSE: Thank you for your comment. We have discussed the feasibility of the methodology and described it in sufficient detail in our response to “6. Review Comments to the Author.”

4. Have the authors described where all data underlying the findings will be made available when the study is complete?

Reviewer #1: Yes

5. Is the manuscript presented in an intelligible fashion and written in standard English?

Reviewer #1: Yes

6. Review Comments to the Author

You may also provide optional suggestions and comments to authors that they might find helpful in planning their study.

Reviewer #1:

The research question needs to be clearly defined.

RESPONSE: We agree with your comment. Therefore, we have added the following sentence (lines 110–113).

Our systematic review and meta-analysis’ research question is: What enhances learning satisfaction and the efficacy of simulation-based infection prevention/control education, compared to classical education, among students who are enrolled in medical, nursing, rehabilitation, and other related courses?

Literature search and study selection: Just two bibliographic databases were included. Typically, three bibliographic databases should be used. In order to secure proper basis for evidence-based research, it is essential to perform a broad search that includes as many studies as possible that meet the inclusion and exclusion criteria.

RESPONSE: Thank you for your suggestion. Accordingly, we added one more database—Scopus. Therefore, we extended the search period from 1990 (January) to 2022 (September). Details regarding this have been added to manuscript (lines 50–51, 114–115, and 158).

Inclusion criteria/ Types of participants: I think the inclusion of studies has biases. It is not comprehensive. Graduate students not included. However, many studies conducted on graduate students with simulation lab. Also, why medicine students and pharmacy students not included, this could yield misleading results.

RESPONSE: Thank you for your suggestion. We agree with you. Therefore, we have added the following sentence (lines 135–138).

The included studies will comprise participants who are undergraduate and graduate students enrolled in medical and healthcare-related occupational courses (medicine, dentistry, nursing, physical therapy, occupational therapy, pharmacy, and other healthcare-related fields).

Guidelines for reporting meta-analyses of RCTs were not presented clearly.

RESPONSE: We agree with your assessment. Accordingly, we have revised the manuscript (lines 200–207 and 213–217).

7. PLOS authors have the option to publish the peer review history of their article (what does this mean?). If published, this will include your full peer review and any attached files.

Do you want your identity to be public for this peer review? For information about this choice, including consent withdrawal, please see our Privacy Policy.

Reviewer #1: Yes: Ala'a Fawwaz Dalky

---

## [Editor Report · Decision Letter 1]

17 Oct 2022

Protocol for educational programs on infection prevention/control for medical and healthcare student: A systematic review and meta-analysis

PONE-D-22-19321R1

Dear Dr. Tashiro,

We’re pleased to inform you that your manuscript has been judged scientifically suitable for publication and will be formally accepted for publication once it meets all outstanding technical requirements.

Kind regards,

Sawsan Abuhammad

Academic Editor

PLOS ONE

Additional Editor Comments (optional):

Congrats!! The paper is accepted!
---

## [Editor Report · Acceptance letter]

20 Oct 2022

PONE-D-22-19321R1 

Protocol for educational programs on infection prevention/control for medical and healthcare student: A systematic review and meta-analysis 

Dear Dr. Tashiro:

I'm pleased to inform you that your manuscript has been deemed suitable for publication in PLOS ONE. Congratulations! Your manuscript is now with our production department. 

Kind regards, 

on behalf of

Dr. Sawsan Abuhammad 

Academic Editor

PLOS ONE